# Design and Testing of a Computer Security Layer for the LIN Bus [note 1]

**DOI:** 10.3390/s22186901

**Published:** 2022-09-13

**Authors:** Felipe Páez, Héctor Kaschel

**Affiliations:** Electrical Engineering Department, Faculty of Engineering, University of Santiago of Chile (USACH), Av. Ecuador 3519, Estación Central, Santiago 9170124, Chile

**Keywords:** computer security, in-vehicle networking, LIN, encryption, HMAC, microcontroller, PSoC

## Abstract

Most modern vehicles are connected to the internet via cellular networks for navigation, assistance, etc. via their onboard computer, which can also provide onboard Wi-Fi and Bluetooth services. The main in-vehicle communication buses (CAN, LIN, FlexRay) converge at the vehicle’s onboard computer and offer no computer security features to protect the communication between nodes, thus being highly vulnerable to local and remote cyberattacks which target the onboard computer and/or the vehicle’s electronic control units through the aforementioned buses. To date, several computer security proposals for CAN and FlexRay buses have been published; a formal computer security proposal for the LIN bus communications has not been presented. So, we researched possible security mechanisms suitable for this bus’s particularities, tested those mechanisms in microcontroller and PSoC hardware, and developed a prototype LIN network using PSoC nodes programmed with computer security features. This work presents a novel combination of encryption and a hash-based message authentication code (HMAC) scheme with replay attack rejection for the LIN communications. The obtained results are promising and show the feasibility of the implementation of an LIN network with real-time computer security protection.

## 1. Introduction

The lack of computer security in vehicles’ internal wired communications (CAN, LIN, FlexRay buses), combined with security flaws in the complex onboard computer’s operating system and user interface, constitutes a real threat to the safety of the occupants by means of remote attacks to this computer, which can spread through these buses to the vehicle’s Electronic Control Units (ECUs). The ECUs control and monitor a wide range of vehicle systems such as the engine, brakes, tire pressure, door and window control, etc. This has been an area of concern for several years among researchers, some of whom have even been able to remotely manipulate certain critical systems of a (physically unmodified) Jeep vehicle since 2014 [1].

We developed a particular interest in LIN bus’s computer security flaws and researched for a possible solution: adding a computer security layer to protect the messages’ confidentiality, integrity, and authenticity in real-time. This additional security layer could be implemented by software in each node’s microcontroller firmware, as our tests proved.

In our first paper about the subject, “Towards a Robust Computer Security Layer for the LIN Bus” [2], we summarized several published computer security proposals for CAN and FlexRay buses, together with a description of known mechanisms for computer attacks on vehicles and possible consequences. We also collected a set of proposed and practical attacks targeting the LIN nodes, highlighting the fact that, to date, no computer security proposal has been presented for the LIN bus. We outlined the main features that would need to be implemented to provide computer security for this bus, such as encryption and authentication, within the old LIN standard’s data width limitations and node response time.

In our second paper about the subject, “A Proposal for Data Authentication, Data Integrity and Replay Attack Rejection for the LIN Bus” [3], we proposed and tested the usage of a hash-based message authentication code (HMAC), together with a timestamp-based mechanism, to provide a robust level of integrity protection and authentication for exchanged data between LIN’s master and slave nodes, and also for protection against replay attacks. We carried out additional hardware tests about this subject to further establish the selected HMAC function (Blake2s); these updated results and analyses are presented here.

In the present article, we detail our search for an adequate encryption algorithm (cipher) for LIN (considering the LIN frame’s inherent limitations) by means of hardware encryption performance testing in order to choose an algorithm.

In addition, we detail our design and testing of a prototype LIN network, implementing a subset of frame types and using PSoC development hardware and real LIN transceivers in order to test the proposed secure communication mechanisms in a more real-life scenario.

We consider this development of a computer security layer for the LIN bus as a major contribution to: (a) the confidentiality and integrity protection of the data exchanged between nodes, and (b) the safety of the vehicle’s occupants. A computer attack over the vehicle’s inner networks can have a variety of consequences, such as erroneous driving decisions based on altered or old data, the remote manipulation of doors and windows or other critical mechanisms such as brakes, etc.

The remainder of this document is structured as follows: Section 2 summarizes the published proposed computer security systems for CAN and FlexRay and the research on LIN bus cybersecurity; Section 3 presents an overview of the LIN bus and then presents a general cybersecurity proposal compatible with the physical layer and the current LIN data format. Section 4 summarizes the encryption mechanism selected, along with hardware performance testing, as does Section 5, which summarizes the selected hash-based authentication proposal and its performance testing. Section 6 presents the design and implementation of a prototype LIN network for testing purposes; finally, Section 7 presents the conclusions and future lines of work for this proposal.

## 2. Related Work

Several computer security proposals have been published in recent years, mainly based on encryption and authentication, for the in-vehicle CAN and FlexRay networks in order to stop or mitigate dangerous computer attacks on the vehicle’s communications. We felt that a review of such proposals could be useful for developing a proper one for the LIN bus; we proceed to summarize these proposals in chronological order.

Vektor Informatik GmbH (2014) presented [4] the development and implementation of a cybersecurity layer for the CAN-FD bus, based on AES encryption. J.A. Bruton (2014) proposed [5] a security system using block encryption, HMAC, and authentication for the CAN bus. Woo et al. (2014) proposed [6] a security layer for CAN, using dynamic key generation, authentication, 32-bit truncated MAC, and real-time encryption based on AES-128 encryption. Spaan (2016) presented [7] a summary on attacks on commercial vehicle models, encryption options, model key exchange, authentication, etc. Fassak et al. (2017) presented [8] a secure protocol to authenticate ECUs on the CAN bus and establish cryptographic session keys between these units based on elliptic curve cryptography (ECC), with real-time communication using symmetric cryptography. Siddiqui et al. (2017) proposed [9] a secure framework for authentication and encrypted end-to-end communication for ECUs, using ECC and AES-128 and implementing FPGA. Liu et al. (2018) presented [10] the design, implementation and experimental results of a FlexRay link layer security scheme based on AES-128 encryption and HMAC authentication based on SHA-1. Alam (2018) proposed [11] the use of symmetric key ECC-based Public Key Encryption (PKE) to ensure confidentiality and the use of digital signatures to ensure integrity and authenticity in in-vehicle networks; he proposed the adoption of an identity-based access control in the master ECU and the use of blockchain technology. Püllen et al. (2019) proposed [12] a model to exploit the optional dual channel mode of FlexRay, providing authentication through MACs, which is compatible with previous versions of the protocol; several methods of message delivery and security mechanisms were discussed. Zhang et al. (2020) presented [13] a CAN security evaluation tool (“CANsec”) that simulates malicious attacks and supports several attack vectors and tested it in a Ford vehicle. The same year, Lee et al. [14] proposed a custom Ethernet/Flexray gateway implemented in FPGA with a “Security Data Transmission Mechanism” to provide computer security to the data exchanged in the gateway. Luo et al. (2021) introduced a proposal for lightweight frame authentication for the CAN bus, tested on CAN and microcontroller hardware. Jadidbonab et al. (2022) proposed [15] a system based on SoC and FPGA technologies in order to add cybersecurity features for the CAN bus in Connected and Autonomous Vehicles (CAVs).

Regarding the LIN bus and its lack of computer security, there appears not to be much published research on the subject, nor is there currently a complete security proposal that we are aware of. In 2017, Takahashi et al. [16] presented a study on vulnerabilities and computer attacks on the LIN bus. In 2018, Ernst et al. [17] presented a study on the security vulnerabilities of the LIN bus, exposing problems such as the implementation of cryptography, the handling of the keys, attacks by means of sleep commands to the nodes, and intrusion detection; a significant conclusion of the authors is that the LIN bus is easier to attack than other buses such as CAN and FlexRay due to its physical layer. In an extensive article (2020), El-Rewini et al. [18] presented up-to-date research that exposes vulnerabilities in both automotive wired networks (CAN, LIN, FlexRay, MOST, etc.) and wireless networks (WiFi, cellular, ZigBee, VANET, etc.).

As a conclusion of this section, it can be stated that the CAN and FlexRay buses have received much attention and the interest of researchers in the computer security field, with several published cybersecurity proposals for both standards (mostly CAN), while the LIN bus has been somewhat neglected in this respect. We felt that current microcontroller technology could allow for a real-time computer security layer for the LIN bus, and the experimental results confirmed that it is feasible.

## 3. Securing Communications over the LIN Bus

### 3.1. LIN Bus Overview

In brief, LIN (Local Interconnect Network) [19] is a serial, master–slave communications protocol (Figure 1) specifically designed for its application in vehicles and featuring error-free communication between various electronic components in the vehicle. Despite its relative age, LIN continues to be widely used in all types of vehicles (over 700+ million nodes were installed in 2020 [20]), which, in recent years, have experienced a notable increase in electronic components for the most diverse tasks, such as the control and supervision of the engine, brakes, and energy and the management of body electronics (windows, seats, air conditioning, door security, etc.). In addition, the LIN bus is also being used by other industries, such as industrial automation (equipment manufacturing, metal-working machines, etc.). The main features of the bus are as follows:Single master node, up to 15 slave nodes; there is no arbitration. This results in a deterministic network with almost zero risk of collisions between nodes.Broadcast communication, bidirectional, half-duplex, one wire, up to 20 kb/s with a bus length of 40 m.Variable data length to be transported, up to 8 bytes.Detection of faulty nodes.Guaranteed latency times by scheduling the master node.Synchronized broadcast reception.Use of synchronization preamble allows for the use of nodes without quartz crystals or ultra-precise time bases.Use of checksum to detect frame errors.Ease of use as a subsystem in hierarchical networks.Physical layer based on ISO–9141.Nominal operating voltage: 12 VDC, 30 VDC max.

The standard LIN message frame is shown in Figure 2. This frame is divided into the header (generated by the master node) and response (the slave node’s response, or even the master node responding to itself for data transmission to slave nodes). The response’s checksum is eight bits in size.

There are several types of LIN frames: unconditional, triggered by events, sporadic, diagnostic, user-defined, and reserved. Communication is managed by the master node, which publishes the headers in the bus and waits for the responses from the slaves in a constant loop with very precise timing (based on a time schedule); communication allows data to be read from slave nodes (for example, originating from sensors connected to them) and to also drive actuators by writing data to the slaves.

The latest version (2.2A) of the LIN standard does not feature any computer security provision, which exposes the network to attacks such as those presented in [2]. The node’s response time (latency) should not exceed 20 ms; any secure LIN communication to be designed must be able to be processed within that timeframe. Security mechanisms such as message encryption and authentication are processor-intensive (and, in some cases, memory-consuming) tasks, out of reach of most of the eight-bit microcontrollers employed in LIN nodes. Another significant constraint is the size of the node response’s data field (the message that we aim to secure): a maximum of eight bytes (sixty-four bits). This will be discussed in the following paragraphs.

### 3.2. Proposed Secure LIN Communication Mechanism

The added computer security features are to be implemented by developing LIN-compatible firmware and testing it on prototype nodes, using regular microcontrollers and SoPC technology.

The goals for our secure LIN communication proposal are as follows: (1) to provide data confidentiality; (2) to provide data authentication; (3) to provide data integrity, and (4) to reject replay attacks.

The first goal is covered by means of encryption, which is covered in the next section of this article. The remaining three goals are covered by means of message authentication and are also summarized in this paper by combining the original LIN data to be transmitted with an original internal node timestamp mechanism detailed in [3].

The LIN standard allows up to eight bytes to be transmitted within the data field (Figure 3 and Figure 4) at one time. This proposal aims to be compatible as much as possible with the current standard, so the data securing will be implemented exclusively on the response’s data field.

This raises a problem: it is impossible to fit eight bytes of unencrypted LIN data in fewer than eight encrypted data bytes, and a message authentication code (MAC) needs a minimum of four bytes to be of practical security (we want to use a full eight-byte MAC for added security). Since there is no LIN “superframe” that could fit these 16 bytes to be sent, we propose two interleaved response frames in sequence, as shown in Figure 4: a 64-bit encrypted block, followed by the 64-bit HMAC.

The communicating master and slave nodes must keep internal track of which response (HMAC data or encrypted data) is being requested/transmitted, which is not a difficult task to implement. Only after both responses have been received and checked can the data received be considered valid. We will first discuss the encryption-related portion of the system. A similar scheme can be used for master-to-slave data frames (LIN sporadic frames).

## 4. Encryption Alternatives and Testing

### 4.1. Cryptosystems: Overview and Requirements for This Development

As previously established, the confidentiality of the exchanged data between LIN nodes is achieved by means of real-time encryption and decryption, so an adversary with unauthorized access to the bus cannot obtain the original data (plaintext) from the encrypted data (ciphertext) in a practical manner. A crucial point is the fact that designing a cryptosystem (or cipher system) for this proposal was pointless, since there are a number of available, well-designed, and thoroughly tested cryptosystems with very different algorithms, key and block sizes, etc. So, the next logical step was to establish the requirements for a possible cryptosystem to be used in this proposal, as follows:It must be able to produce a 64-bit ciphertext (the encrypted LIN data) from a 64-bit plaintext (the original LIN data).The encryption and decryption processing times must be as short as possible, ideally in the range of a few milliseconds or less on embedded hardware. To a lesser extent, RAM memory usage should be also small.It must have not been broken by practical, full-round attacks.

Considering the second requirement and the limitations of the node hardware (regarding the processing power, which is much less than that available in a personal computer), we had to make a major decision: the use of an asymmetric or symmetric cryptosystem.

An asymmetric cryptosystem, such as ECC [21], RSA [22], etc., provides a good level of security at the expense of much processing power because of the discrete logarithm concept involved. This kind of cryptosystem relies on a public key distributed among the participants, used to encrypt data, along with a secret private key required for decryption. This also implies long processing times, even in the range of several seconds in common eight-bit microcontroller hardware, and is therefore impractical for real-time encryption and decryption in such systems.A symmetric cryptosystem, such as AES [23], Blowfish [24,25], Simon/Speck [26,27], etc. is based on a single private key known by the participants. This kind of encryption is very fast to compute and less taxing on the system’s processing unit than public key cryptosytems, and it is therefore adequate for real-time encryption and decryption.

Because of the above, in communication systems, asymmetric cryptography is more suited for exchanging secret keys before a nominal communication flow is established. From that point on, real-time encryption and decryption of the transmitted/received data is performed via symmetric cryptography. So, assuming that the proposed secure LIN nodes could have a factory-preprogrammed, shared secret key, we opted for symmetric cryptography. Having established this, we had to choose between a stream or block cipher; the former operates on a bit-by-bit or byte-by-byte basis in a streaming manner, whereas the latter operates over fixed-size data blocks. Because the LIN response data field will be used to its full capacity in fixed 64-bit data blocks, a block cipher was the obvious choice, (Figure 5; where *b* = 64; *c* = 64).

### 4.2. Considered Ciphers

Three block ciphers were considered and performance-tested, taking into account the aforementioned requirements (most prominently the 64-bit block size):*Blowfish* is a symmetric key cryptosystem designed in 1993; to date, no effective cryptanalysis has been performed. It is a fast block cipher, except when changing the key; a new key requires the preprocessing equivalent of encrypting around 4 kB of plaintext, which is relatively slow compared to other block ciphers. Additionally, this amount of memory puts it beyond the reach of very basic microcontrollers and embedded systems based on them; however, this is not a problem for modern 32-bit microcontrollers that usually feature at least 8 kB of RAM. There are also eight-bit microcontrollers with 8 kB of RAM, such as the ATmega1280/2560, which could perform this encryption in reasonable time. Blowfish was one of the first secure block ciphers that is patent-free and therefore freely available for anyone to use; this feature has contributed to its popularity in crypto software. Blowfish’s key sizes range from 32 to 448 bits; its block size is 64 bits. Plaintext less than 64 bits must be zero-padded to the required 64-bit size.*Tiny Encryption Algorithm* (TEA) [28,29] is a block cipher that stands out for its simplicity of description and implementation; it was introduced in 1994. This cipher is not subject to any patent. TEA works with two 32-bit unsigned integers (two halves of a 64-bit data block) and uses a 128-bit key and 32 suggested rounds. It has an extremely simple key schedule, shuffling the subkeys in exactly the same way for each cycle. TEA was initially designed to be a small implementation algorithm in terms of the memory required [30]. As the basic operations are very simple, TEA is also considered as a very high-speed encryption algorithm and as suitable for embedded systems. Two further variants, XTEA (1997) and XXTEA (1998), were intended to improve the security of TEA but introduced new weaknesses; as a result, TEA is still used.*Speck* is a family of lightweight block ciphers that was released publicly by the NSA in June 2013. Speck is optimized for performance in software implementations and is an add-rotate-XOR (“ARX”) cipher that does not substitution boxes, unlike Blowfish. The origins of these ciphers date back to 2011, when the NSA began to work on them, anticipating that some US federal government agencies would need encryption that would work well across a diverse collection of IoT devices while maintaining an acceptable level of security [27]. The authors of the cipher consider it to be very fast compared to other block ciphers (including AES) and have shown the feasibility of implementation [27] in FPGAs, ASICs, and 8-, 16-, and 32-bit microcontrollers; in all cases, both the encryption speed and processing resources used are Speck’s selling points. On ARM architecture, Speck is practically three times faster than AES.

None of the three considered ciphers have been successfully attacked under real-life scenarios, which is a notable achievement, especially considering the age of Blowfish and TEA. Considering that fact, the most important criteria for choosing one of the aforementioned cryptosystems were the execution time for encryption/decryption and, to a lesser extent, the amount of RAM required. Now, the hardware platforms used for testing will be described.

### 4.3. Hardware Platforms for Testing

**General considerations**. We felt that it was important to conduct tests on real hardware, representative of LIN nodes, in terms of processing time, because computer security mechanisms are processor-intensive tasks not suited for low-range microcontrollers. In this sense, three RISC CPU architectures were considered as microcontroller brains in order to be able to execute the tests of process time, memory usage, and the general feasibility of encryption/decryption: 8-bit AVR architecture (well-known due to its massive use on 8-bit Arduino platforms), 8-bit PIC18F architecture (also widely used), and 32-bit ARM architecture (widely used in cell phones, tablets, printers, hard drives, pendrives, USB in general, etc.).

**AVR-based platform**. The Microchip AVR architecture is available in a range of microcontrollers with various memory capacities, integrated peripherals, numbers of pins, etc. The CPU is eight-bit, RISC-type, and very efficient in terms of execution, reaching up to 16 MIPS @ a 16 MHz clock frequency. The AVR architecture was introduced in the early 2000s and is still widely used in millions of devices at a very competitive cost. As a case study, the ATmega328P [31] and ATmega2560 microcontrollers were chosen, both being part of the AVR family. Regarding the 328 micro, it features 32 kB of program memory, 2 kB SRAM, three timers/counters, and several internal peripherals. Regarding the 2560 micro, it has the same architecture and CPU speed as the 328 micro, but it is equipped with more peripherals and port pins (irrelevant for this development), more program memory (256 KB), and RAM (8 KB). For testing purposes, the relevant elements of these micros are the CPU and its clock frequency and the required size of RAM and program memories. The internal timers allow for the measurement of the execution time of the cipher algorithms with a precision of 1 μs. The tests were carried out in the Arduino development environment (Figure 6) using the Nano variant that incorporates the 328 chip. In certain test scenarios, the program memory and RAM of this chip turned out to be insufficient; for this case, the ATMega2560 chip, in Arduino Mega Core form, was used.

**Microchip PIC18-based platform**. This family of microcontrollers is considered to be low-to-middle-range, offering a set of features similar to AVR micros (RISC CPU, memory, peripherals, etc.) but with a lower execution efficiency; PIC18 family micros require four clock pulses to execute their instructions, reaching a maximum of 10 MIPS @ 40 MHz CPU clock; note that the previously discussed AVR micros go up to 16 MIPS @ 16 MHz (ATmega2560) and 20 MIPS @ 20 MHz (ATmega328). However, these PIC devices are still widely used despite their relative age and thus provide an interesting point of comparison, especially to the previously discussed eight-bit AVR family. The particular model to be used will be the PIC18F4680 [32] in 40-pin DIP format, the main reason being its ample program memory (64 kB), which allows it to accommodate encryption, hash, and HMAC algorithms that take up a lot of instructions and tables, such as BLAKE2 or RIPEMD. Other relevant features for this work are its 3KB SRAM memory and a variety of timers which allow for precise measuring of the encryption processing time. The hardware board used is a PICDEM Plus model (Figure 7); a 4 MHz crystal was used, in conjunction with the micro’s internal PLL (fixed 4x multiplier); this drives the CPU to work at 16 MHz, on par with the other micros for testing purposes.

**ARM-based platform**. The ARM CPU microarchitecture is ubiquitous today as the main processor of countless devices and embedded systems. The use of 32-bit ALUs and registers and a micro-design optimized for high performance in MIPS v/s power consumption makes this architecture ideal for portable devices. It is necessary to note the dramatic difference in the capabilities and processing speed of an ARM CPU compared to an eight-bit CPU at the same clock frequency, as will be seen later when comparing the encryption/decryption, hashing, and HMAC-processing times. This architecture is usually implemented in microcontrollers or other system-on-a-chip equivalents. The tests were carried out on the ARM-CortexM3 architecture—in particular, on the Cypress CY8C5888LTI-LP097 chip [33] in USB module form (Figure 8); it is a programmable system-on-a-chip (PSoC) and essentially consists of a 32-bit ARM CPU core, an FPGA area interconnected with the CPU, 64 kB SRAM memory, 256 kB of Flash program memory, 2 kB EEPROM, and other non-digital and hybrid blocks such as ADCs, DACs, OpAmps, etc. What really matters for the tests is the CPU section, the memory required, and the use of internal timer blocks to measure the execution speed of the code. For development purposes, the IDE provided by Cypress (PSOC Creator 4.2) was used, which allows for the creation of peripherals in the FPGA area and programming in C language (by means of the GNU gcc compiler) for the ARM CPU. The CPU processing time was quantified by means of a hardware counter with a resolution of 1
μs.

### 4.4. Encryption Performance Results

Before proceeding with the tests, the following Table 1 shows the key sizes and number of rounds executed for each tested cryptosystem. In all cases, the plaintext and cyphertexts are 64 bits in size. Note that the number of rounds (algorithm’s iterations) is a key parameter, since various theoretical attacks on these cryptosystems base their results on execution with fewer rounds than those recommended, or standardized, for these cryptosystems, i.e., not full-round usage. We used full-round tests for each cryptosystem.

Both the AVR and PIC18 microcontrollers were tested at a 16 MHz CPU clock, while the PSoC ARM core was tested at 16 MHz (for direct comparison with the AVR and PIC18 micros at the same speed) and 64 MHz. We feel that modern technology makes it perfectly feasible to use a 64+ MHz chip in an LIN node at a reasonable cost, should that be the case. The tests’ results are presented in Table 2.

Regarding Table 2, we developed firmware for the microcontrollers using C and C++ languages, written using Arduino IDE (for AVR micros), Microchip MPLAB (for PIC18F), and PSoC Creator (for Cypress PSoC). The PIC18F chip could not run the Blowfish64 test because of its RAM size of 3 kB. It is also noteworthy that, in practical terms, the decryption times are almost the same as the encryption ones (as will be shown in Section 6), so Table 2 is a valid reference for both encryption and decryption performance. The PIC18 architecture was the worst in terms of general execution speed, although the TEA performance was more than acceptable in this case. Blowfish needs about 4 to 5 kB of RAM because of the pre-computed s-boxes, a relatively high amount for a basic microcontroller; this narrowed the decision to TEA or Speck (Blowfish did not offer special qualities over these ciphers). We preliminary chose the Speck64 cipher because of its outstanding performance in ARM architecture (and acceptable AVR performance); it is also a contemporary cryptosystem with little RAM memory consumption. However, since the proposed security system must also provide authentication using MAC, a significantly time-consuming process in a microcontroller, it was necessary to perform the MAC hardware tests in order to confirm Speck64 as the chosen cipher for our proposal (because of the combined execution times for both the encryption and MAC computing).

Regarding side-channel attacks, there is some research related to the Speck cipher, including a very interesting one using transfer-learning techniques, “Side Channel Analysis of SPECK Based on Transfer Learning” [34], focusing on the XOR operations used by the cipher. However, the study itself acknowledges that “Aiming at the problem that the lightweight algorithm is difficult to crack due to the lack of nonlinear operation”, and a successful key extraction was not achieved. Side channel attacks are also difficult to execute in real time and require large quantities of data traffic in order to achieve its goal. Finally, to the best of our knowledge, no successful side-channel attack has been performed on a full-round Speck algorithm (which is precisely the mode that we used in our proposal).

## 5. MAC Alternatives for the LIN Bus

### 5.1. Summary

In our previous article [3], we established the need for a MAC in order to provide authentication and data integrity for the LIN data to be transported within the bus. The MAC is transmitted after the encrypted data, allowing the receiver to check the veracity of the message. We decanted for hash-based MAC (HMAC), a technique for obtaining an MAC by means of an underlying hash function. As a quick reminder, a hash function is a computed mathematical transformation applied to a block of binary data of arbitrary size in order to obtain a fixed-size digest or “hash”; a change of one or more bits in the original data block dramatically changes the result of the hash (the so-called avalanche effect).

We also proposed an in-node 32-bit time counter mechanism to protect against replay attacks, which is to be combined with the 64-bit LIN data before applying HMAC computing to the combined 96-bit data (the 64-bit LIN data concatenated with the 32-bit time counter).

A brief recap of the timer counter mechanism is as follows: A 32-bit counter register is embedded into each prototype node; each node’s register increases at the same rate. This counter starts as zero every time the whole LIN network is simultaneously powered up. Before a response is sent, the original, unencrypted LIN data (8 bytes) is concatenated with the 4 bytes of the counter register, and the whole 12-byte array is HMAC’ed into 8 bytes (the 2nd LIN response). When the master receives the HMAC response, it computes its own HMAC by first concatenating the eight decrypted bytes with the master’s own four-byte counter. A fresh message will yield identical received and computed HMACs that are treated as valid. However, replayed messages will not be valid because its HMAC, computed by the slave using an old counter value, will not match the master’s computed HMAC using its updated counter.

In the same article, we tested the hardware execution times for several hash functions and their corresponding HMACs, tentatively choosing HMAC-Blake2s, considering some crucial advantages such as fast processing in microcontroller hardware and the built-in keyed hash offered by Blake2s.

In the present article, we expanded the scope of the hash and HMAC tests by including the PIC18F microcontroller previously discussed and updated the result tables, allowing for further analysis and commentary.

### 5.2. Updated Test Results

We used the same development tools, hardware platforms, and CPU clocks previously used for encryption tests; as before, the hash functions and HMACs were coded using C/C++ language and tailored for each platform and its development software. The following are the updated execution test results for the hash and HMAC functions considered.

Regarding Table 3, it can be seen that RIPEMD-160 offers the lowest processing times in ARM architecture; the results of BLAKE2s are quite good, making it feasible to use it on eight-bit AVR micros. SHA is the least suitable in terms of the required processing time, especially on the eight-bit platforms. All hash functions perform very slowly on PIC18F hardware; the execution times are well past the 20 ms mark. A significant advantage of BLAKE2s is that it does not require the truncation of the results, unlike the other tested hashes, which return more than 64 bits (in this case, the recommended procedure by the NIST [35] is to keep the n most significant bits of the hash). What will finally be used for the proposed system will be an HMAC based on a specific hash; therefore, Table 4 presents the execution times for the previously discussed HMACs.

As seen on Table 4, the HMAC processing times are several times greater than the corresponding hash functions, except for Blake2s (which requires roughly double the time). As previously mentioned, the shortest execution time for the security mechanisms is desired in an LIN node; therefore, in terms of authentication, the chosen HMAC type to be used in this security system proposal was BLAKE2s.

Considering ARM’s much shorter times required for the encryption, decryption, and HMAC mechanisms, the development continued on ARM architecture, also considering its wide availability and low cost of implementation.

## 6. Prototype LIN Network Design and Testing

### 6.1. Introduction

The tests performed in Section 5 and Section 6 proved the feasibility of running the security mechanisms in real-time using common microcontroller hardware. These tests were performed in the microcontrollers as isolated units, using internal data buffers with no communication. However, we wanted to go further and test those mechanisms with communication functionality in a real prototype LIN network as a testbed.

The LIN bus is a fairly complex standard to implement (despite its apparent simplicity when compared to CAN or FlexRay), especially at the network and higher layers, considering the communication planning (scheduling), the subscription and publication mechanism of the nodes’ master and slave tasks, etc. The diagnostic frames (ID 60,61) make the issue more complex. Considering that this proposal “retrofits” computer security elements in the data bytes of the response frame, it is necessary to point out that the scope of the tests does not consider LIN nodes programmed with the complete standard, since the core of the experiment is the verification of the proposed security mechanisms. That being said, the following elements are involved:

A real LIN physical layer using specialized transceivers, combined with microcontrollers or equivalent SoC, to implement nodes in the network.One master node, one “normal” slave node, and one “attacker” slave node (Figure 9).The communication will use the security mechanisms previously chosen: Speck64 and HMAC Blake2s; in both cases, a pre-shared 128-bit key is used.The nodes’ prototype firmware includes the following:
-An SCI block emulated with a UART and additional FPGA logic.-LIN header publishing and response reception in two discrete buffers in the master node for decryption and HMAC verification, respectively; because of the very nature of the development and testing procedure, the first header will be manually triggered, the results will be displayed on each node’s LCD screen, and no time scheduling is needed nor implemented.-The usage of an LIN-enhanced checksum.-A 64-bit LIN response, generated by the slave nodes, according to the particular scenario being tested. This data will be the result of encryption and HMAC (normal slave) or plaintext data (attacking slave) in alternate response fashion.The 32-bit counter register of each node protecting against replay attacks, as discussed, was set to increase every 250 ms, providing an ample communication window for testing purposes.

### 6.2. Testbed Hardware Design

**LIN node microcontroller.** As previously established, the 32-bit ARM architecture was chosen as the CPU for testing purposes; specifically, the Cypress CY8CKIT-059 module chip previously discussed (32-bit ARM CPU, 256 kB Flash program memory, 64 kB SRAM, and a built-in FPGA area). These resources are somewhat oversized for the proposed application, but at the same, time they provide enormous flexibility for hardware design. In particular, to send and receive LIN frames, a UART block will be used, combined with available logic in the FPGA area of the Cypress chip; such blocks will be assigned to arbitrary physical pins of the CY8CKIT module, which in turn will be connected to an LIN transceiver. The Cypress chip’s CPU will be powered by a 16 MHz clock, a medium-to-low speed for the capabilities of this architecture, which is more representative of a real LIN node. According to the performance tables presented, such a clock speed allows for very fast processing of the characteristics of the proposed security system. The programming was carried out in the Cypress IDE, using C language to implement the necessary parts of the LIN standard for communication between the nodes of the proposed network, along with the security mechanisms.

In order to ease the hardware and software development, a special board (Figure 10a), adding an LCD screen, buttons, and other peripherals to the Cypress module, was used for each LIN prototype node. The screen allowed for in-node, real-time information display such as LIN data, execution times, etc. and proved to be very useful for this development.

**LIN transceiver.** The LIN node’s microcontroller must incorporate a serial communication interface (SCI), very similar to a UART, which is responsible for sending and receiving serial data to and from the bus. However, the physical connection to the bus itself is through the LIN transceiver, which is connected to the LIN wiring, that is, to the physical layer of the bus (which has strict electrical specifications, protection against ESD, bit timing, etc.). For this particular case, the eight-pin TJA1021 LIN transceiver chip [36] has been chosen in the form of a breakout module (Figure 10b) powered by a 12 VDC laboratory power source. The transceiver is driven by the Cypress chip by means of the TX and RX signals; no additional control or enable signal is used.

Each of the prototype nodes is composed of a development board as a base unit (programmed and debugged via USB) plus the LIN transceiver, as shown in Figure 11. Note that the same hardware is used for the three prototype LIN nodes; each one’s role in the network is determined by its custom firmware.

In addition to the C language programming, tailored for each of the nodes according to their function in the prototype network, SCI blocks were created for each in the FPGA area of the Cypress chip, combining a UART with logic gates, registers, and edge detectors, in order to simulate each node’s LIN SCI block. Figure 12 shows the modified UART block used in the slave nodes. The RX OR gate avoids self-reception when transmitting the response, the edge detectors (using a 1 MHz sampling rate) allow for the master node’s header frame detection, and the TX OR gate allows for logic ‘1’ forcing on the transmitting line. The CPU programming has complete control over these elements.

We devised and tested two LIN network test scenarios: without and with the attacking node.

These scenarios allowed us to contrast the theoretical and practical behavior of our LIN-compatible programming, the security mechanisms, and the general viability of our proposal.

We did not set up a scenario to solely test the attack replay rejection, because the HMAC with a counter register is built in to each prototype node’s specific programming and works seamlessly.

### 6.3. Test Scenario 1: Unconditional Frame Transmission, No Attack

This scenario tests the response generated by the slave node (normal) when detecting the header from the master node, which is the most common LIN communication frame. The attacker node does not participate. This scenario is useful for testing the communication itself (logical and physical layers), the base programming of the master and normal slave nodes, and the security mechanisms’ function and performance.

The key parameters for the test are as follows:Both the master and slave nodes share a 128-bit key, which is equal to $030201000B0A0908131211101B1A1918. This is the same key used for encryption, decryption, and HMAC computing.The slave node has been assigned an arbitrary LIN ID = $31 (which translates to an LIN PID = $B1).The slave node holds 32 bits of plaintext LIN data, equal to $13941761, an arbitrary value. These data could represent a number of automotive sensors attached to the LIN slave. Internally, this 32-bit word will be zero-padded to 64 bits before applying the encryption and HMAC mechanisms.These data will be transmitted to the master in two LIN responses (corresponding to two successive headers with the same PID, as previously discussed; Figure 4): the encrypted data go first, followed by the HMAC data.The test begins when the master node’s hardware pushbutton is pressed: immediately after, the first header is published in the prototype LIN bus.

Figure 13 shows the initial screens displayed by both the master and slave (normal) nodes, waiting for the communication to begin. The data to be transmitted by the slave node (“LINslv_N”) are displayed on its own screen.

When the first header is published, the slave node first checks the PID, which must match its own. In such case, the encrypted data are calculated and sent as a standard LIN response. After reception, the master publishes an identical header, and the slave node calculates the HMAC and sends it to the master, again as a standard LIN response. Then, on the screen, the slave shows the execution times (μs) for both the encryption (“CyphT=”) and HMAC (“HMACT=”) calculations on each of the two passes (Figure 14). These times are obtained using an internal hardware counter. If the PID does not match, no response is sent by the slave node.

On the master’s side, two internal RAM buffers are used to separately receive the encrypted and HMAC data before any processing; if both buffers have a valid LIN checksum, the decryption is executed. Both the encrypted data (“Rx.Cyph=”) and the decrypted LIN data are shown in the LCD (“Decyph=”) (Figure 15). All data are presented in hexadecimal format; in this case, the decrypted data are the same as the original LIN slave’s data shown in Figure 13b.

By pressing the same button on the master node’s hardware, the HMAC result screen is shown; here, both the received HMAC data (“Rx.HMAC=”) and the calculated HMAC data (“CalHMAC=”) are displayed (Figure 16). A mismatch between the two implies that a forged, inauthentic message was transmitted to the master node.

The next display shows the HMAC comparison between the received and computed HMAC (Figure 17). If both HMACs are exactly the same, the data are authentic and valid.

The next and final display shows the master’s processing times required by both the decryption processing (“DcrypT=”), HMAC calculations (“HMACT=”) and the total time (“TotalT=”, in microseconds), required by the master node to obtain the full LIN data from the slave (Figure 18). The displayed time of 14.1 ms includes:The time required for the two LIN frames to be transported, taking into account the 19.2 kbps communication speed.The time required by the slave node for communication and security mechanisms’ processing.The time required by the master node for communication and security mechanisms’ processing.The generation and validation of checksums (this is a very small portion of the total time, in the range of microseconds).

After this screen, pressing the reset button on the master node board returns it to the initial state, and the test can be repeated. No such reset is needed in the slave node board.

### 6.4. Test Scenario 2: Unconditional Frame Transmission with Attacking Slave

In this scenario, the attacking node will intentionally corrupt the normal slave’s LIN response by publishing its own in the bus at the same time. The attack will affect the first frame sent to the master (i.e., the encrypted data published in the bus by the normal slave) by simultaneously publishing a 64-bit “attack word”. We developed a small C-language program in order to generate random alternatives of a 64-bit attack data that, when combined with the normal data in the bus, would yield the same checksum as the normal data, thus fooling the LIN checksum validation. Example words of attack data obtained with this program are $14EAB6333269C9EF, $1192EDB98A10AAB9, and $44B441E51683D9AD. We used the first one for our case study.

The second frame, corresponding to the HMAC data published by the normal slave, will not be attacked. Figure 19a shows the attacking slave’s initial state (identified as “LINslv Att” and showing the attack data—in this case, $14EAB6333269C9EF). Figure 19b shows the attacking node’s display after sending the response (no encryption nor HMAC is performed by this slave in this scenario).

The result of the attack on the master node is shown in Figure 20. As a result of the LIN physical layer, the master received the equivalent of a bitwise AND operation between the data bits simultaneously published by both slaves; the result of this AND operation yields a “correct” checksum value ($8B), the same checksum value of the encrypted data without an attack, thus passing the first, standard level of verification in the master. It is important to highlight that decryption itself cannot detect the alteration of the original data ($13941761), but the HMAC computation allows the master to identify the altered data ($1B762139, Figure 20a) as not authentic (Figure 20c); notice the distinct received and computed HMAC values; Figure 20b.

## 7. Conclusions and Future Work

Through this research, development, and testing, we have presented a robust computer security layer for the LIN bus that is built from scratch, compatible with its physical and link layer, computationally feasible to implement, and capable of real-time processing, with the complete communication scheme (involving two header-response frames) carrying the information between nodes well below 20 ms, as shown in Figure 18. It has also been built using flexible microcontroller technology, which is common, compact, and widely available.

We think that this proposal is a valuable contribution to the security of internal communications in vehicles’ LIN networks, and, by extension, it contributes to the security of the occupants by preventing the exploitation of the LIN bus’s data traffic, the altering of critical sensor information, and the injection of old data, which could lead to very serious consequences.

The encryption and decryption mechanisms mitigate LIN data spying in a practical manner, while the HMAC mechanism allows the receiver to discard malicious, forged messages sent from the attacking node; it also allows for the discarding of old, replayed messages.

Regarding the performance tests and the prototype LIN network, the usage of various development boards was crucial for the programming and testing; it even allows for easy in situ display of the communication status and other information. Beyond the somewhat bulky development and test hardware used, real LIN nodes based on this technology can be built in a very compact fashion by just using the Cypress chip, the transceiver chip, and some additional circuitry.

As future developments, we would like to add dynamic key exchange between the nodes at the startup time, which can be achieved by public key cryptography (such as ECC). This would prevent the need to use a pre-stored, factory key in all the nodes. As a starting point, we have tested a 192-bit ECC encryption routine in similar ARM hardware, obtaining the result in about 70 ms, which we consider a reasonable startup time for each node in the LIN network.

We also would like to implement more types of LIN frames by means of additional programming for each of the prototype nodes.

Regarding the PID, being essentially a single byte after the sync bits, we did not modify its form or function in our prototype LIN network. A computer-security protected PID would involve adding bytes to the header (such as MAC), and we did not want to deviate that much from the current LIN standard; however, that could also be considered as a further development.

Finally, we would like to research alternative ways of transporting the encrypted and HMAC data, but that inevitably implies more deviation from the LIN standard. One way could be to assign two correlative IDs to one slave, even and odd, such as $30 and $31, or $1A and $1B. Instead of two identical ID headers published by the master, as explained, the slave could deliver the encrypted data when responding to its even ID and the HMAC data when responding to its odd ID. While this idea is completely feasible using the same hardware and slightly different programming, the number of available nodes per LIN network would be cut in half. Another method could be the implementation of the previously mentioned “superframe”, i.e., a 16-byte data field in the slave’s LIN response, instead of the current 8. That way, the eight bytes of encrypted data and the eight bytes of HMAC data could be sent to the master in one frame, avoiding the interleaved sequence. While this is theoretically possible, it would require additional research on its impact on both the checksum calculation and the physical layer.

## Figures and Tables

**Figure 1 sensors-22-06901-f001:**
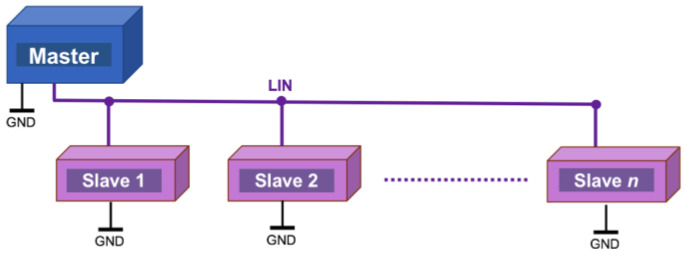
LIN topology, supporting up to 15 slave nodes.

**Figure 2 sensors-22-06901-f002:**
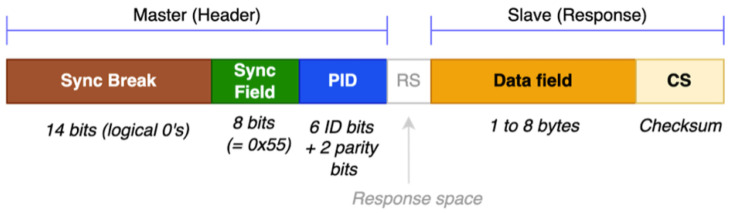
Details of the current, unsecured LIN message frame.

**Figure 3 sensors-22-06901-f003:**
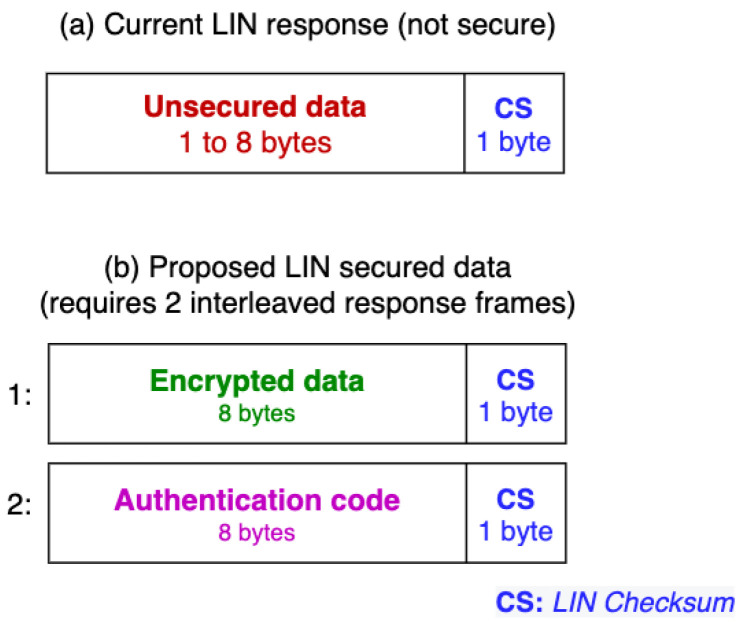
(**a**) Current usage of the response’s data field; (**b**) Proposed usage of this field, with secured data.

**Figure 4 sensors-22-06901-f004:**
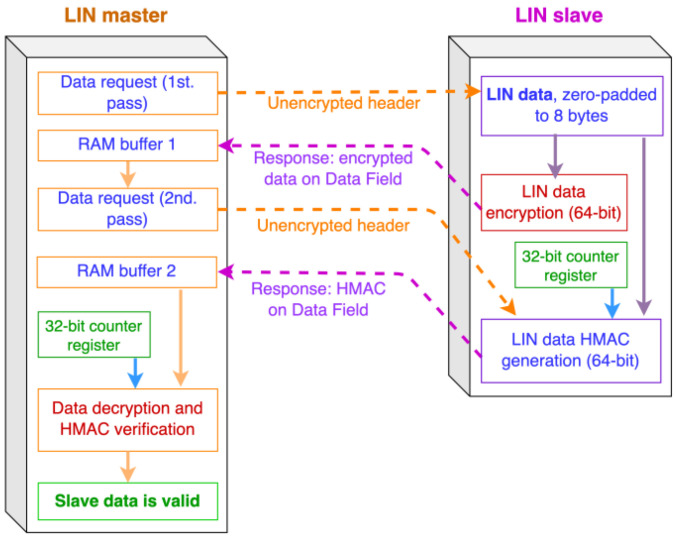
Proposed LIN communication sequence using 64-bit encryption and HMAC on alternate frames.

**Figure 5 sensors-22-06901-f005:**
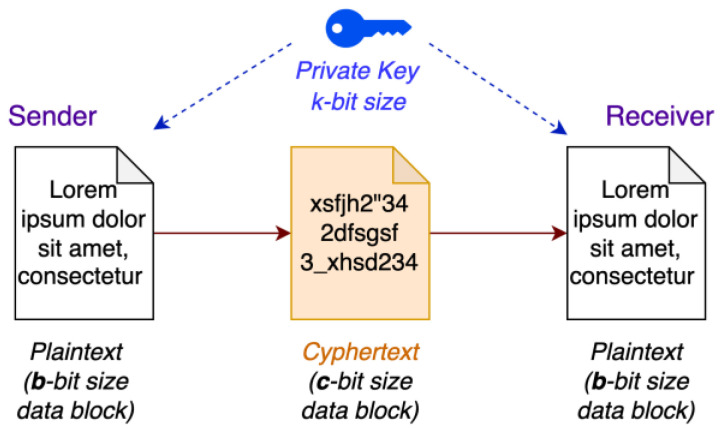
Block cipher with a symmetric key.

**Figure 6 sensors-22-06901-f006:**
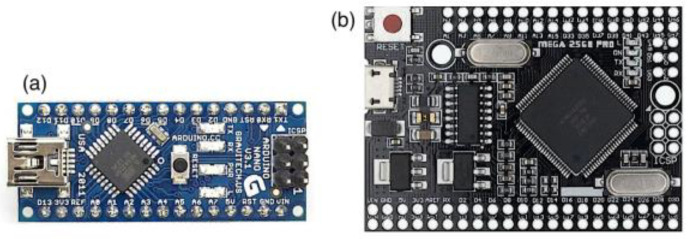
Platforms used for AVR testing: (**a**) Arduino Nano, (**b**) Arduino Mega Core.

**Figure 7 sensors-22-06901-f007:**
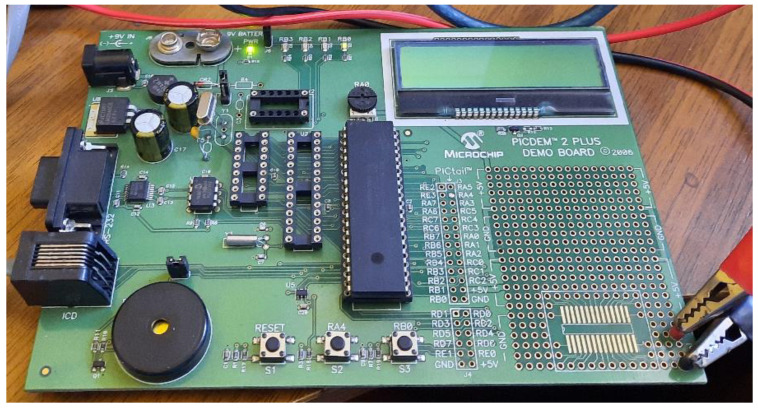
PICDEM development platform used for PIC18F performance testing.

**Figure 8 sensors-22-06901-f008:**
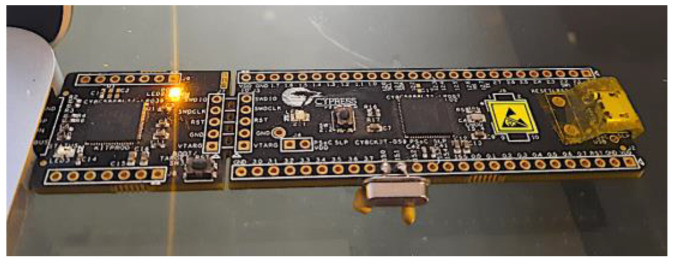
Cypress development USB stick used for performance testing.

**Figure 9 sensors-22-06901-f009:**
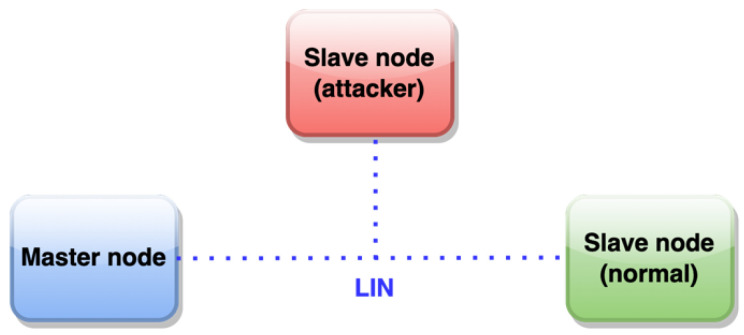
LIN prototype network testbed.

**Figure 10 sensors-22-06901-f010:**
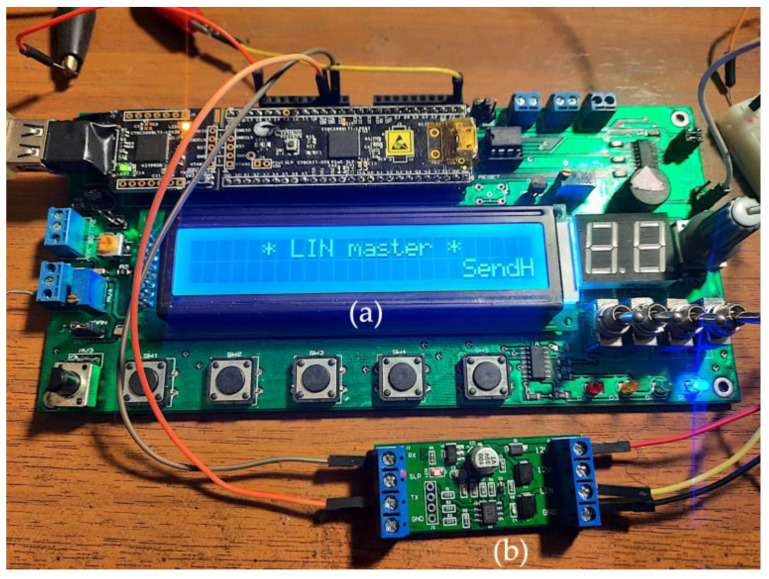
Prototype LIN node composed of a (**a**) development board and (**b**) LIN transceiver.

**Figure 11 sensors-22-06901-f011:**
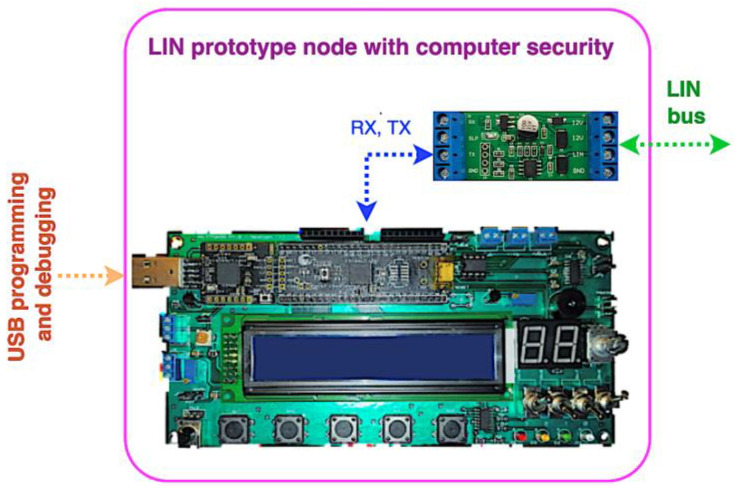
Prototype LIN node structure for network testing.

**Figure 12 sensors-22-06901-f012:**
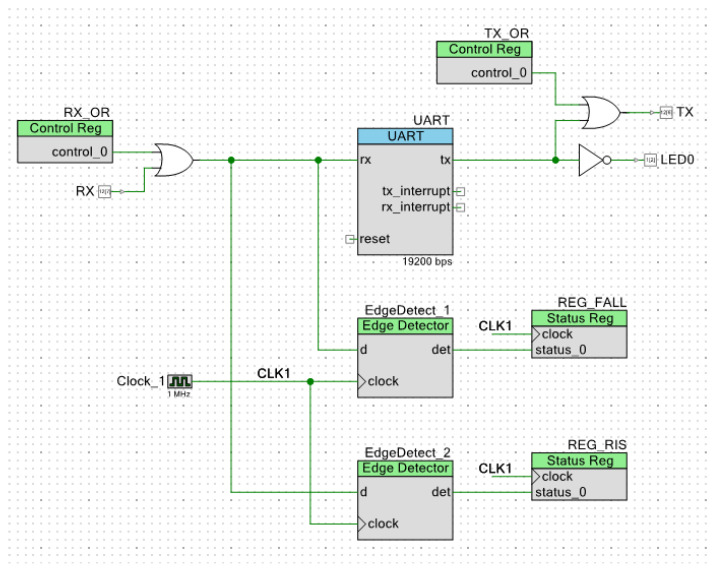
Modified UART block created in the slave nodes’ FPGA area.

**Figure 13 sensors-22-06901-f013:**
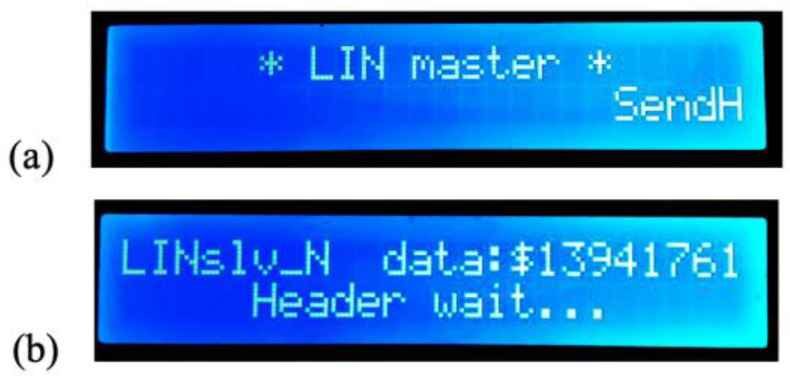
Initial states displayed after reset or power-on: (**a**) master node, (**b**) normal slave node.

**Figure 14 sensors-22-06901-f014:**
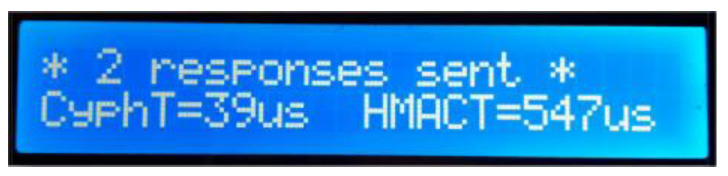
Normal slave node’s screen status after sending the two interleaved LIN responses.

**Figure 15 sensors-22-06901-f015:**
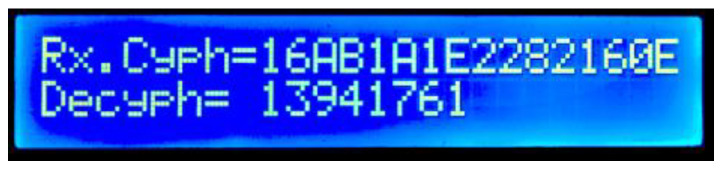
Master node’s decryption result display.

**Figure 16 sensors-22-06901-f016:**
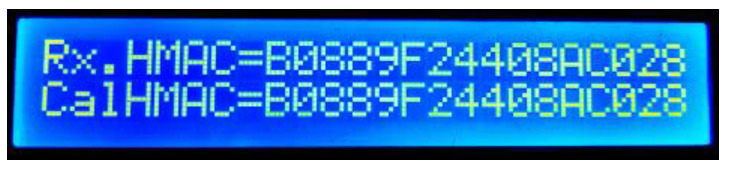
Master node’s first HMAC display.

**Figure 17 sensors-22-06901-f017:**
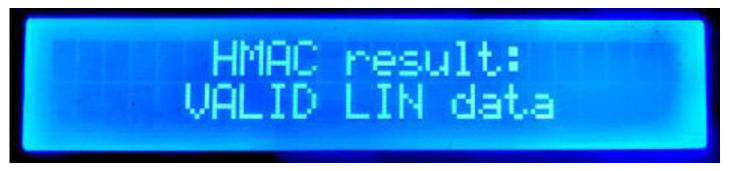
Master node’s second HMAC display.

**Figure 18 sensors-22-06901-f018:**
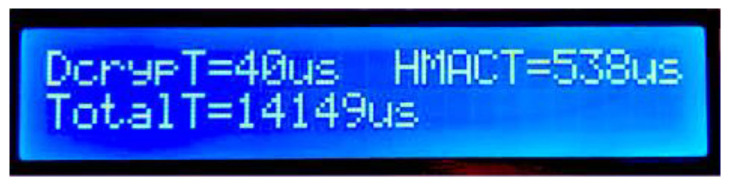
Master node’s decryption, HMAC computing and total time display.

**Figure 19 sensors-22-06901-f019:**
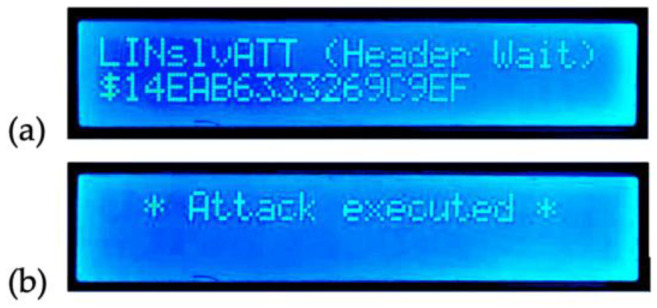
Attacking slave node’s screen displays; (**a**) initial state, (**b**) result after response sent.

**Figure 20 sensors-22-06901-f020:**
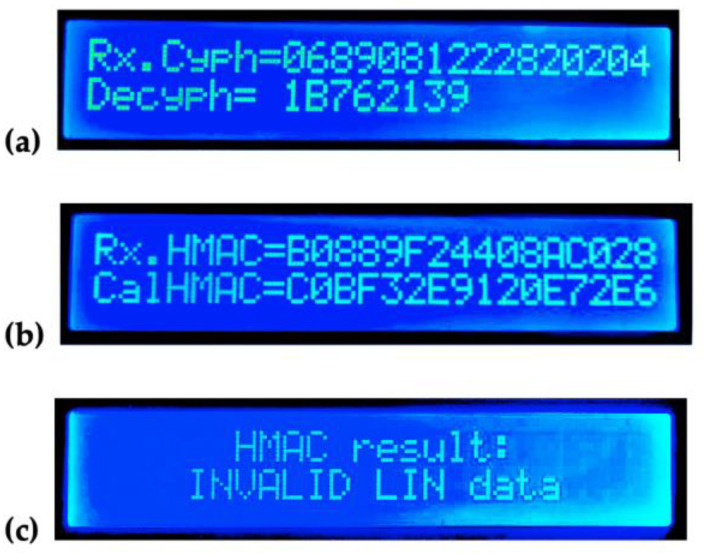
Master node’s state after the attacked response; (**a**) decryption result; (**b**) HMAC result; (**c**) LIN data validity result.

**Table 1 sensors-22-06901-t001:** Test Parameters for the Considered Cryptosystems.

Cryptosystem	Key Size	Rounds
Blowfish64	56 bits	16
TEA64	128 bits	32
Speck64	128 bits	27

**Table 2 sensors-22-06901-t002:** CPU Processing Times for Encryption.

Criptosystem	AVR @16 MHz	PIC18F @16 Mhz	ARM @16 MHz	ARM @64 MHz
Blowfish64	124 μs	N/A	50 μs	13 μs
TEA64	376 μs	689 μs	44 μs	11 μs
Speck64	1.28 ms	8.14 ms	56 μs	14 μs

**Table 3 sensors-22-06901-t003:** Processing Times for the Considered Hash Functions (less is better).

Hash Function	AVR @16 MHz	PIC18F @16 Mhz	ARM @16 MHz	ARM @64 MHz
RIPEMD-160	5.87 ms	29.1 ms	256 μs	95 μs
BLAKE2s	3.41 ms	31.8 ms	293 μs	96 μs
SHA-224	11.1 ms	57.5 ms	434 μs	112 μs

**Table 4 sensors-22-06901-t004:** Processing Times for the Considered HMAC Functions (less is better).

HMAC Type	AVR @16 MHz	PIC18F @16 Mhz	ARM @16 MHz	ARM @64 MHz
HMAC-RIPEMD160	23.1 ms	70.6 ms	923 μs	350 μs
HMAC-BLAKE2s	6.78 ms	61.6 ms	539 μs	181 μs
HMAC-SHA224	44.3 ms	230.0 ms	930 μs	490 μs

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
