# Peer review of "Design and Testing of a Computer Security Layer for the LIN Bus†"

_sensors, 2022, doi:10.3390/s22186901_

Round 1

Reviewer 1 Report

This paper proposes a cryptographic protocol for securing LIN communications.  According to its design, it uses symmetric encryption and  a message authentication code to meet the following security objectives:

1.    Provide data confidentiality.  

2.    Provide data authentication.

3.    Provide data integrity,

4.    Reject replay attacks. 

According to its description, the master and the slave share a secret key (128 bits). The protocol consists of 4 messages interchanged between the master and the slave

1.    Unencrypted header

2.    Response: Encrypted data in the Data field (by using a proper symmetric encryption scheme)

3.    Unencrypted header

4.    Response: HMAC in the data field (by using a proper message authentication code)

To this reviewer, there is no clarity on how this approach can reject replay attacks as claimed (objective 4).

Let us assume an attacking slave (as test scenario 2 – section 6.4) that passively collects previous messages interchanged between the normal slave and the master in a protocol round (Particularly, message 2 and message 4). That attacking slave may resend collected messages 2 and 4 in a future protocol round between the master and the normal slave. Messages 2 and 4 are valid. Hence a replay attack would be feasible. This motivates the following questions:

1.    How is the freshness of a message guaranteed?

a.    What data is being HMAC’ed by a slave?

b.    How PID is treated?

On the other hand,

Since a key is shared between the slave and sender, would it be possible to retrieve such a key through a side-channel attack? There is no mention of it in the article. This reviewer thinks adding a subsection discussing side-channel attacks relevant.

Author Response

Please see the attachment, thanks.

Reviewer 2 Report

According to my understanding this research discusses the networking threats, challenges and future effects for computer security in detail. Some comments given here to address

No were mentioned about the parameters value reported in the manuscript.

- related work need to be detailed in a way to find their limitations in the tabular format. And mention which of the limitations/ limitations is to be handled by your work. 

Abstract is poor. it can be extended by discussing the results on proposed work

-  Reorganize the introduction, trying to explain every word of the title.

- Please highlight the contribution clearly in the introduction

- in the proposed model, if we increase the number of slave nodes to 20 , what will happen

- how did you do the hardware testing for this? explain neatly by adding one paragraph atleast

- What are the evaluations used for the verification of results?
- Major contribution was not clearly mentioned in the conclusion part.

Author Response

Please see the attachment, thanks.

Round 2

Reviewer 1 Report

The authors have addressed this reviewer's concerns satisfactorily.